# Molecular and Clinical Implications of Somatostatin Receptor Profile and Somatostatin Analogues Treatment in Oral Cavity Squamous Cell Carcinoma

**DOI:** 10.3390/cancers13194828

**Published:** 2021-09-27

**Authors:** Alba Sanjuan-Sanjuan, Emilia Alors-Perez, Marina Sanchez-Frías, Alicia Dean-Ferrer, Manuel D. Gahete, Susana Heredero-Jung, Raúl M. Luque

**Affiliations:** 1Maimonides Institute for Biomedical Research of Córdoba (IMIBIC), E-14004 Cordoba, Spain; b12alpee@uco.es (E.A.-P.); marinasanchezfrias@gmail.com (M.S.-F.); adeanferrer@yahoo.es (A.D.-F.); bc2gaorm@uco.es (M.D.G.); susana_heredero@yahoo.es (S.H.-J.); 2University Hospital Reina Sofía (HURS), E-14004 Cordoba, Spain; 3Oral and Maxillofacial Surgery Department, University Hospital Reina Sofía (HURS), E-14004 Cordoba, Spain; 4Department of Cell Biology, Physiology, and Immunology, University of Córdoba, E-14004 Cordoba, Spain; 5Centro de Investigación Biomédica en Red de Fisiopatología de la Obesidad y Nutrición, (CIBERobn), E-14004 Cordoba, Spain; 6Anatomical Pathology Service, Maimonides Institute for Biomedical Research of Córdoba (IMIBIC), University Hospital Reina Sofía (HURS), E-14004 Cordoba, Spain

**Keywords:** somatostatin receptors, oral cavity cancer, head and neck, biomarkers, somatostatin analogues, therapeutic tool

## Abstract

**Simple Summary:**

The treatment of oral squamous cell carcinoma (OSCC) represents a significant problem worldwide. Among cancers with the highest incidence, OSCC renders one of the worst prognoses. Therefore, novel prognostic biomarkers and therapeutic tools to tackle OSCC are urgently needed. Somatostatin-analogues (SSA) are an invaluable therapeutic option in the treatment of several cancers. We aimed to determine the expression levels of all somatostatin-receptors (SSTs) in OSCC, compared to adjacent healthy control tissues, to analyze the relationship of SSTs expression with key clinical and histopathological data, and to explore the direct in vitro effect of different SSAs on OSCC cancer cells. Our findings highlight a potential role of SST_2_ as a good prognostic biomarker for recurrence and metastasis in OSCC and unveil that SSA exerts antitumoral effects on OSCC cells, providing a relevant clinical conclusion, which should be soon tested for their use in humans.

**Abstract:**

Oral squamous cell carcinoma (OSCC) incidence has increased by 50% over the last decade. Unfortunately, surgery and adjuvant radiotherapy and chemotherapy are still the mainstream modality of treatment, underscoring the need for alternative therapies. Somatostatin-analogues (SSA) are efficacious and safe treatments for a variety of tumors, but the presence of somatostatin-receptors (SSTs) and pharmacological effects of SSA on OSCC are poorly known. In this study, we demonstrated that SST_2_ and SST_3_ levels were significantly higher in OSCC, compared to adjacent healthy control tissues. SST_2_ expression was associated with less regional metastasis and a lower recurrence rate. Moreover, SST_2_ was elevated in OSCC and associated with histopathological good prognosis factors, such as high peritumoral inflammation, smaller depth of invasion, and expansive vs. infiltrative front of tumor invasion. Importantly, treatment with different SSA (octreotide, lanreotide, and pasireotide) significantly reduced cell-proliferation in OSCC primary cell cultures. Altogether, this study demonstrated that SST_2_ is overexpressed in OSCC vs. healthy tissues and could represent a novel prognostic biomarker, since its expression is associated with tumors that show better prognostic factors and less recurrent rate. Moreover, our data unveil clear antitumoral effects of SSAs on OSCC, opening new avenues to explore their potential as targeting therapy to OSCC.

## 1. Introduction

Oral squamous cell carcinoma (OSCC) continues to be an aggressive disease and a worldwide challenge [1]. Depending on risk factors, surgery and adjuvant radiotherapy +/− chemotherapy remain the mainstream modality of treatment for local or advanced disease [2]. Despite all efforts, OSCC five-year survival-rate still accounts for 60% of patients [2,3,4]. Therefore, novel prognosis biomarkers and therapeutic targets for OSCC are urgently needed.

Neuroendocrine differentiation has been found in some tumors not considered to be of neuroendocrine origin, including SCC of the lung and esophagus [5], and more recently in the head and neck region [6,7,8]. In this regard, somatostatin (SST) is a well-known inhibitory neuropeptide that is produced in different central and systemic locations [6,9,10]. SST inhibitory actions are mediated through their so-called SST receptors (SSTs), which are widely distributed in normal and tumor tissues, and regulate, among other activities, cell proliferation, differentiation, and angiogenesis in many tumor types [11]. This property allows them to play a valuable role in tumor imaging (SST-scintigraphy or octreotide scan) [12]. In this sense, tumors cells typically express more than one SST-subtype, being the most frequently expressed SST_2_ subtype, and thus the most important target [13,14,15,16]. Consequently, synthetic SST analogues (SSAs) represent an attractive therapeutic target to treat the SST-positive tumor pathologies controlling hormone hypersecretion and tumor growth [17,18,19].

Our current understanding of the presence of SSTs on OSCC is quite scarce and unclear. These limited studies have shown that the relative immunohistochemical expression of some SST-subtypes are altered in malignant lesions in the larynx, compared to benign regions [7,8], and in tumor samples of the head and neck area, compared to normal oropharyngeal mucosa specimens (obtained during uvulopalatopharyngoplasty) from other patients [6]. However, to the best of our knowledge, no molecular analyses have been performed to analyze quantitatively, in a side-by-side manner, the expression levels (copy number) of all SSTs subtypes in OSCC samples, compared to healthy tissue (control; within the same patient) using quantitative PCR. Moreover, to date, the direct effects of SSAs on primary OSCC human cell cultures have not been tested.

Therefore, based on the information mentioned above, the objectives of this study were: (1) to quantitatively analyze the expression profile of SSTs in OSCC vs. adjacent healthy tissues obtained within the same patient in a well-characterized cohort of patients; (2) to assess the putative in vivo association between the expression of all SSTs in the tumor and relevant clinical/histopathological data parameters (stage, histological grade, tumor invasion, presence of metastasis, recurrence, overall survival, etc.); and (3) to explore and compare, side-by-side, the direct antitumor effects of different SSAs (octreotide, lanreotide, and pasireotide) in primary OSCC human cell cultures.

## 2. Materials and Methods

### 2.1. Patients Data and Samples Collection

The Ethics Committee of the Reina Sofia University Hospital (Cordoba, Spain) approved the study, which was conducted in accordance with the Declaration of Helsinki and national and international guidelines and approved by the Ethics Committee of the Reina Sofia University Hospital (Cordoba, Spain, Approval # 70180004). Written informed consent was obtained from all the patients. A prospective observational case–control study was performed with 37 patients diagnosed with oral cavity SCC (lip, tongue, floor of the mouth, buccal mucosa, upper and lower gingiva, retromolar trigone, and hard palate). Patients were followed up for at least 24 months after surgery. Clinical variables were obtained from the clinical chart. Specifically, stage (I/II/III/IV), histological grade (G1, G2, G3), tumor pT stage (pT1, pT2, pT3, pT4), cervical metastasis or pN (pN (pN0, pN1, pN2a, pN2b, pN2c, pN3), depth of invasion (DOI), perineural (PNI) or lymphatic/vascular invasion (LVI), peritumoral inflammation (PTI), pattern of tumor invasion, lymph nodes size and extranodular extension (ENE+) were recorded. Some variables, such as Stage, DOI, pT, pN, and PTI were divided in subcategories (pN × 4: pN0, pN1, pN2, pN3; DOI × 3:DOI × 3, <5 mm, 5–10 mm, >10 mm) or in dichotomous categories (Stage × 2: I + II/III + IV; pT × 2:pT1pT2/pT3 + pT4; pN × 2: pN0 + pN1/pN2 + pN3; pN − (pN0)/pN + (pN1, pN2, pN3); PTI × 2: absent + low/moderate + severe) to allow better analysis. Disease overall survival (OS) and OS rate at 24 months were calculated. OS was defined as the period between the diagnosis and death. Disease-free survival (DFS) was defined as the period between the primary surgery and the first recurrence, the last examination, or death. Three patients, who died before six months due to perioperative complications, were classified as “lost data” for recurrence analysis. Overall recurrence rate (RR), local recurrence, regional recurrence, local and regional combined, and distant metastasis were calculated.

OSCC tumor tissue samples (case) were obtained from the surgical specimen after resection. Healthy adjacent tissue samples (control) were obtained within the same patient from the buccal mucosa with a distance from the tumor greater than 2 cm. Then, both specimens were immediately deposited in cold culture medium and transported to the laboratory. The control sample and a fragment of the tumor tissue were frozen at −80 °C for subsequent RNA isolation, retrotranscription, and expression analysis by quantitative real-time PCR (see below). When possible, the remaining tumor tissue was used to perform cell cultures (see below). It should be mentioned that the tissue sample was always obtained in a safe and ethical manner and did not interfere with the pathologist’s work in any case.

### 2.2. RNA Isolation and Retrotranscription (RT)

Total RNA from all samples was extracted at the same time using the RNase-Free DNase Set (Qiagen, Limburg, The Netherlands), according to manufacturer instructions, as previously reported [20,21]. The amount of RNA recovered and its purity was determined using the Nanodrop One Spectrophotometer (Thermo Fisher Scientific, Madrid, Spain). One μg of total RNA was retrotranscribed to cDNA with the First-Strand Synthesis kit (MRI Fermentas, Hanover, MD, USA) using random hexamer primers in a 20 µL volume, as previously reported [22].

### 2.3. Quantitative Real-Time PCR (qPCR)

qPCR reactions were performed using the Brilliant III SYBR Green QPCR Master Mix (Stratagene, La Jolla, CA, USA) in the Stratagene Mx3000p system and specific (and validated) primers for each transcript of interest, as previously reported [17]. For each reaction, 10 µL of SYBER Green, 8.4 µL of Water, 0.3 µL of Forward and Reverse Primers (10 µM) and 1µL of the sample (50 ng of cDNA) were used. The qPCR was made according to the following program: 1 cycle at 95 °C for 3 min, 40 cycles of denaturing (95 °C for 20 s) and annealing/extension (61 °C for 20 s), and a last cycle, where final PCR products were subjected to graded temperature-dependent dissociation (55 °C to 95 °C increasing 0.5 °C/30 s) to verify that only one product was amplified. Specifically, human transcripts for SST receptors (*SST_1_*, *SST_2_*, *SST_3_*, *SST_4_*, *SST_5_*) were used, as previously reported [17]. To control for variations in the amount of RNA used in the reverse transcription reaction and the efficiency of the reverse transcription reaction, the expression level (copy number) of each SST transcript was adjusted with a normalization factor calculated from actin-beta, hypoxanthine-guanine phosphoribosyltransferase 1, and glyceraldehyde 3-phosphate dehydrogenase expression levels (used as housekeeping genes), as reported previously [23]. In this sense, samples were run in the same plate, against a standard curve for each of the transcripts analyzed to estimate absolute mRNA copy number of each transcript and a No-RT sample as the negative control. Additionally, products were run on a 2% agarose gel and stained with RedSafe (iNtRON, ABC Scientific, Glendale) to confirm that only one band was amplified, and no primer dimers were formed. An aliquot of the PCR products was then purified using the MinElute PCR Purification kit (Qiagen) and the purified PCR products were then sequenced to confirm target specificity.

### 2.4. Primary OSCC Cell Culture

OSCC tissues were placed after surgery in sterile cold PBS 1× (Omega Scientific, Tarzana EEUU) with 1% antibiotic-antimycotic solution and immediately dispersed into single cells under sterile conditions by enzymatic and mechanical disruption and cultured onto tissue culture plates in serum-containing medium. Specifically, samples were minced into 1–2 mm^3^ pieces with a sterile scalpel and washed twice with PBS 1×. Then, pieces were incubated in culture medium supplement with Dispase (Invitrogen, Carlsbad, CA, USA) and Collagenase I (Invitrogen) for 30–60 min shaking at 37 °C (up to 2 h). The dispersed cell suspension was centrifuged and washed twice. Cell incubation continued with 5 mL of 0.05% Trypsin-EDTA (Sigma-Aldrich, Madrid, Spain) for 5 min at 37 °C, following of incubation with 15 mL of DNase I (Promega, Madrid, Spain) for 5 min at 37 °C, with siliconized pipette agitation every 5 min. Cells were filtered through a nylon gauze of 130-µm mesh (to avoid fibroblast contamination) and dissociated into individual cells by repeated smooth tipped siliconized glass Pasteur pipette aspiration. Finally, RBC lysis (BioLegend, London, UK) treatment was used to eliminate possible red cell contaminations. Cell number and viability (always higher than 95%) were determined by the trypan blue dye exclusion method (American Type Culture Collection, Manassas, VA, USA) in a Neubauer Chamber. Cells suspension were seeded in RPMI 1640 (ThermoFisher Scientific) medium supplemented with 10% fetal bovine serum (FBS), 1% antibiotic-antimycotic, and 2 mM L-glutamine in plates previously coated with poly-L-lysine to enhance cell adherence.

### 2.5. Cell Proliferation Assay

Cells were plated in 96-well plates at the density necessary to obtain a ~75% cell confluence in the control groups at the end of the experiment (10.000 cells/well). Twenty-four-hour later, serum-free medium was added for 24 h. Then, cells were incubated for 3 h in 10% Alamar Blue reagent/serum-free medium, and Alamar Blue reduction (basal cell viability) was determined in a FlexStation3 system (Molecular Devices, Sunnyvale, CA, USA) plate reader, exciting at 560 nm and reading at 590 nm. After this, different SSAs [first generation (octreotide and lanreotide; with high-affinity binding to SST_2_ and SST_5_) and second generation (pasireotide; a multireceptor-targeted SST with high affinity for SST_1_, SST_2_, SST_3_, and SST_5_) at 10^−7^ M (dose previously reported to exert the most potent antitumor actions in different endocrine-related tumors [24,25])], and vehicle-treated controls were added to wells (at least 4 wells/treatment) in 5% FBS medium for 24, 48 and 72 h. Alamar Blue reduction was measured every 24 h, as previously reported [26]. All assays were repeated a minimum of three times on independent days.

### 2.6. Statistical and Bioinformatical Analysis

All data are expressed as mean ± SEM. Statistical analysis was performed using SPSS (IBM, New York, NY, USA) and GraphPad Prism (La Jolla, CA, USA). Normality was assessed using Shapiro or Kolmogorov-Smirnov test and by visual inspection of the shape of histograms. We evaluate data heterogeneity of variance using the Kolmogorov-Smirnov test to compare the difference between the means of the gene’s expression levels in tumor tissue and healthy tissues within the same patient. Consequently, parametric (Student t) or nonparametric (Mann-Whitney U) tests were implemented. For differences among two groups, One-Way ANOVA analysis was performed to explore statistical differences.

Survival curves were calculated by Kaplan–Meier analysis, and the log-rank test was used to compare OS and Recurrence according to different variables. Parametric or nonparametric tests were used to analyze the relationship between risk factors, clinical and staging data, histopathological analysis, and SSTs expression levels. In vitro cell proliferation experiments were assessed by multiple comparison tests (one-way ANOVA followed by Dunnet post-hoc test) and performed in a minimum of three independent primary cultures from different patients (at least 4 replicates/treatment per experiment), and results are expressed as percentage of control (vehicle-treated cells; set at 100%). Statistical significance was considered when *p* < 0.05. A trend was considered when *p* < 0.1.

## 3. Results

This study includes the analysis of 37 patients diagnosed with OSCC, 19 men (52%) and 18 women (48%), with a mean age of 64 ± 2-years-old (range 26–86 years). The patient’s follow-up time was 24 to 43 months after surgery. The 2 years overall survival (OS) was 76% with a rate of 20.2 ± 1.3 (range 2–24) months. Our cohort is comprised by 50% of patients with advanced Stages IV, 16% with Stage III, 29% Stage II, and 5% with Stage I; 34% of our patients belonged to pT4 tumors, 23% were pT3, 37% were pT2 and 5% were pT1. The cervical lymph node involvement was positive in 42% with pN1 in 10%, and with pN2 and pN3 both in 16%. The recurrence analysis showed that the overall recurrence rate (RR) was 26% (9/34), the local recurrence was 23% (8/34), the regional was 21% (7/34), and both local and regional combined recurrence was 15% (5/34). The distant metastasis rate in the cohort accounted for 9% (3/34).

### 3.1. Expression of Somatostatin Receptors in the Healthy Oral Cavity and OSCC Tissues

A variable expression level for each of the five SST subtypes was found in OSCC (Figure 1). Specifically, the present work revealed that *SST_1_* is the dominant SST subtype expressed in healthy oral cavity tissues (mean ± SEM: 9408 ± 2737 mRNA copy number), followed by *SST_2_* > *SST_5_* > *SST_4_* > *SST_3_* (5245 ± 999; 4432 ± 1437; 2454 ± 1312; 347 ± 157; respectively). In contrast, this profile was found to be altered in OSCC samples being SST_2_ the dominant SST subtype expressed (mean ± SEM: 24,245 ± 5730 mRNA copy number), followed by *SST_5_* > *SST_4_* > *SST_1_* > *SST_3_* (8698 ± 3561; 7295 ± 4381; 6318 ± 2648; 2171 ± 652, respectively). Thus, when we compared the expression levels between OSCC and healthy samples, we found that in general, the expression of all receptors, except *SST_1_*, was increased in OSCC, compared to healthy adjacent control samples, being this increase statistically significant for *SST_2_* and *SST_3_* (*p* < 0.01 and *p* < 0.001, respectively). No sex differences were found in the expression of SSTs.

### 3.2. In Vivo Association between SST-Subtypes Expression in OSCC with Relevant Clinical and Pathological Parameters

As previously reported [27,28], to perform the relationship between the expression levels of SSTs in OSCC tissues and the different clinical and histopathological data (including the Kaplan Meier curves), we represented the expression levels of SSTs as numerical or categorical (expression level higher (>) or lower (<) median values). It should be noted that given the high number of analyses that were performed, and in order to simplify the representation of these associations, we decided to include only the “*p*” and corresponding “R” values of these analyses in the tables described below.

OS and Recurrence: Our analyses revealed that higher expression of *SST_2_* (the dominant SST subtype expressed in OSCC samples) was related to a lower rate of regional recurrence (*p* = 0.04) and both local and regional recurrence (*p* = 0.02) (Table 1). We also found a trend for significant association (*p* = 0.07) between higher expression of *SST_2_* and lower presence of distant metastasis (Table 1). Moreover, we found that higher expression of *SST_5_* showed a trend for a lower incidence of both local and regional recurrence (*p* = 0.05) (Table 1).

Staging data: The univariate analysis showed that *SST_2_* expression was statistically increased on patients with less cervical nodal disease [pN (*p* = 0.02), pN × 4 (*p* = 0.02), pN × 2 (*p* = 0.05); Table 2].

Patients with pN− vs. pN+ showed higher *SST_2_* expression (*p = 0.03*; Table 2). This was also observed with the categorical analysis where patients with higher SST_2_ expression presented less cervical nodal disease [pN (*p = 0.04*), pN × 4 *(p = 0.05*), pN − /pN + (*p = 0.03*); Table 3]. *SST_2_* had no relationship with pT; however, *SST_3_* expression above the median had a positive correlation with a higher pT and a higher Stage [pN × 2 (*p = 0.03*) and Stage × 2 (*p = 0.04*), respectively; Table 3].

Histopathological data: *SST_5_* expression was statistically increased in G1 tumors (*p = 0.02*; Table 4), which are the well differentiated tumors. Moreover, *SST_1,2,4,5_* expression was statistically increased in OSCC that had an expansive front of tumor invasion, compared to OSCC, with an infiltrative front of tumor invasion (*p = 0.01, p = 0.03, p = 0.03, p < 0.01*, respectively; Table 4). Similarly, our results showed that *SST_1,2,4,5_* were overexpressed in OSCC with uniform tumor invasion edges, compared to the poorly defined ones (*p=0.08*, *p = 0.08*, *p = 0.03*, and *p < 0.01*; Table 3) (*p = 0.01*, *p = 0.03*, *p = 0.03*, and *p < 0.01*, respectively; Table 4).

Our data also revealed that patients with higher *SST_2_* expression more frequently presented tumor depth of invasion (DOI) of 5–10 mm, compared to >10 mm and <5 mm (*p = 0.04*; Table 3). *SST_2_* expression also showed a statistical tendency to be present on tumors that showed a higher peritumoral inflammation reaction [PTI (*p = 0.06*) and PTI × 2 (*p = 0.08*), Table 3].

Finally, we found that the expression of *SST_2_* in OSCC had a negative correlation to the number of positive lymph nodes, the number of lymph nodes with ENE+, and their bigger size (*p < 0.01*, *p = 0.03*, and *p = 0.05*, respectively; Table 5).

### 3.3. Antitumor Actions of First- and Second-Generation Somatostatin Analogues on Patient-Derived Primary Oral Squamous Carcinoma Cell Cultures

In the present study, we also explored the effect of different SSAs (octreotide, lanreotide, and pasireotide) on the proliferation rate of patient-derived primary OSCC cell cultures. Remarkably, our results demonstrated that all SSAs tested (10^−7^ M) significantly reduce the proliferation rate of primary OSCC cell cultures OSCC (Figure 2). Specifically, all SSAs decreased proliferation rate at 24-, 48-, and 72-h of incubation (this inhibition was not statistically significant in the case of octreotide at 24 h and lanreotide at 72-h) (Figure 2).

## 4. Discussion

Oral cancers are among the most common malignant tumors worldwide, wherein more than 90% of all oral malignancies are OSCC, which significantly reduce patients’ quality of life [29]. Importantly, although OSCC is considered a disease of old age, a recent clinical scenario witnesses its increasing incidence among young individuals [30]. In fact, according to recent statistics from the International Agency for Research on Cancer (http://gco.iarc.fr/, accessed on 22 September 2021), the number of OSCC cases that are newly diagnosed each year is very worrying. Therefore, this high incidence, together with the hidden onset, low survival rate, and the limited and inefficient treatments, clearly emphasize the necessity of identifying novel biomarkers for these tumors. These potential biomarkers would help to refine OSCC diagnosis, to better predict their prognosis and tumor behavior, and provide tools to develop novel therapeutic targets.

In this study, we have investigated the expression pattern of all SSTs in parallel using a quantitative PCR method in a group of samples derived from patients with OSCC (tumor vs. adjacent non-tumor tissues and evaluated their potential relationship with key clinical and pathological parameters. To the best of our knowledge, this is the first time that the expression of SST in OSCC has been thoroughly and quantitatively (mRNA copy number) analyzed in a relatively large series of samples. In the present series, we observed a differential SST expression pattern in OSCC tissues (*SST_2_* >> *SST_5_* > *SST_4_* > *SST_1_* > *SST_3_*), compared to their corresponding adjacent non-tumor tissues (*SST_1_* > *SST_2_* > *SST_5_* > *SST_4_* > *SST_3_*). Moreover, we demonstrated the existence of an overall increase in the expression of *SST_2,3,4,5_* in OSCC samples, compared to control tissues, being this overexpression statistically significant for *SST_2_* and *SST_3_* levels. This might be considered an important clinical finding, as the responsiveness of SSAs is critically dependent on the presence of SSTs, and because the treatment with available SSAs (e.g., first generation compounds, octreotide and lanreotide, which preferential bind to *SST_2_*) has become the mainstay of medical therapy for tumor control in neuroendocrine disorders expressing SSTs, such as pituitary and gastroenteropancreatic neuroendocrine tumours [18,19]. In this sense, during the last decade, neuroendocrine differentiation has been found in some tumors not considered to be of neuroendocrine origin, including SCC of the lung, esophagus, larynx, head, and neck [5,6,31], suggesting that the use of neuropeptides analogues (e.g., SSAs) could be used as a potential therapeutic avenue for OSCC.

Our observations compare favorably with previous reports indicating that the expression of different SST-subtypes, including *SST_2_* and *SST_3_*, is consistently increased in other tumors, compared to normal tissues, including human prostate [32,33], pituitary [17,21,34], and neuroendocrine tumors [14,25], among others, and with previous scarce observations, indicating that the head and neck squamous cell carcinoma specimens express different SST-subtypes (mainly *SST_1,2,5_* using semi-quantitative immune-histochemical staining) [6,7,8]. Hence, it seems reasonable to suggest that overexpression of SSTs might be a common cellular/molecular signature across various tumor types and that SSAs may have a therapeutic role in these tumors.

Another relevant finding of our study was that the expression levels of different SSTs, especially *SST_2_* (the dominant SST subtype expressed in OSCC samples), were associated with malignancy features. Firstly, we found a correlation between higher *SST_2_* expression and less regional metastasis. Moreover, *SST_2_* expression also had a negative correlation to the number of positive lymph nodes, the number of lymph nodes with ENE+, and their size, which are well-known risk factors in OSCC for regional recurrence and distant metastasis [35]. Additionally, we observed that the expression of *SST_2_* and *SST_5_* was related to a lower rate of regional and both local and regional recurrence. On the contrary, the expression of *SST_3_* (the SST subtype with lower levels in OSCC samples) was positively correlated with a higher pT and a higher Stage. Therefore, these results show that the main clinical correlations were associated with *SST_2_* expression in oral cavity SCC, a dominant receptor in these tumors that seems to be associated with decreased malignancy features, suggesting a potential prognostic value as metastatic and recurrence biomarker.

Prognostic factors of oral cavity cancer are well known and under continuous review. DOI was recently introduced to the major changes in the last 8th TNM Edition [35]. Our results showed a higher expression of *SST_2_* (*p = 0.04*) in tumors with DOI 5–10 mm (15/37), compared to >10 mm (18/37) or <5 mm (4/37). We believe this last observation might be associated with the small number of pT1 (<5 mm DOI) analyzed in the study, due to the limiting selection criteria. Besides DOI, recent papers have also focused on the impact of tumor budding and the pattern of invasion in recurrence rate [36,37]. We have found that SSTs are related to histopathological factors known to be present in tumors with a less aggressive histopathological behavior, such as an expansive vs. infiltrative front of invasion, lower histopathological grade, or uniform edges, compared to the poorly defined ones. In addition, peritumoral inflammation has been considered a defense mechanism against cancer progression and invasion [38,39,40]. It has been proposed that inflammatory activity, such as an immunological response to the tumor, could be used as a prognostic factor since the lower the inflammatory infiltrate, the greater the risk of regional or distant metastasis. However, the PTI’s role in the prognosis of OSCC is still very controversial. Other studies suggest peritumoral stromal inflammation is more likely to contribute to cancer development [41]. Our results showed that higher PTI had better survival curves and a lower recurrence rate. Furthermore, *SST_2_* expression shows a statistical tendency to be present on tumors that show a higher PTI. Hence, all the data mentioned so far in the present study support the idea that the expression of *SST_2_* might be used as a potential prognostic biomarker in OSCC and provide a scientific rationale for a randomized controlled trial of an SST analogue (especially those that preferentially bind to *SST_2_*) in these tumors.

Thus, given the result obtained in the present study indicating that OSCC tissues express high levels of different SST-subtypes and that the expression of SST (specially *SST_2_*) is associated with relevant clinical and pathological data of OSCC patients, together with the poor outcome in advanced cases, and the marginal survival benefit of toxic chemotherapy regimens, this disease demands testing novel therapeutic strategies such as SSAs. In fact, to the best of our knowledge, our results are the first to demonstrate that OSCC cells are responsive in vitro to first and second generation of SSAs (octreotide, lanreotide, and pasireotide). Interestingly, all SSAs tested equally reduced cell proliferation in OSCC cells; therefore, no evident correspondence was found with the SST profile. Thus, it seems plausible that additional factors, besides the simple abundance of a given SST, critically influence the SSA response in OSCC cells, as has been previously suggested in other tumor pathologies [24].

The present study has some limitations: the limited number of cases analyzed and the short follow-up time for the analysis of the OS, which is usually measured on a 5-years period. Therefore, we will continue increasing the number of samples for future investigations and continue following these patients for a proper analysis of the impact of SSTs on patient´s survival. However, it should be noted that it is known that local or regional events occur within the first two years [42], which was the minimum follow-up time of all our patients.

## 5. Conclusions

In summary, this report assesses the expression levels of all five SST subtypes in OSCC by qPCR, and it is the first series to compare the expression levels of each receptor between OSCC and control (adjacent non-tumor) tissues. Additionally, although the role of SSTs as possible prognostic biomarkers and therapeutic targets in OSCC needs to be further explored, this study strongly suggests that: (1) expression levels of *SST_2_* could be related with less rate of regional recurrence, both local and regional and less incidence of distant metastasis, suggesting that the assessment of SST expression profiles by qPCR may represent an effective screening tool to predict prognosis of OSCC; and, (2) SSAs exert antitumoral effects on OSCC cells, opening new avenues to explore their potential as novel targeting therapy for patients with OSCC.

## Figures and Tables

**Figure 1 cancers-13-04828-f001:**
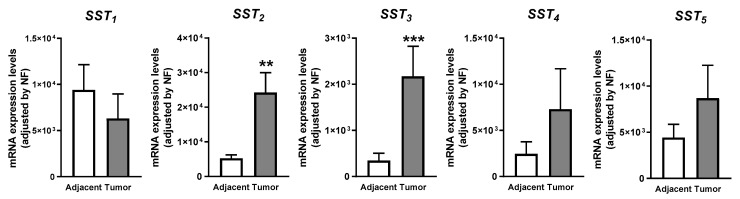
mRNA expression levels of Somatostatin Receptors. mRNA expression levels of SSTs genes were measured by qPCR and adjusted by normalization factor. Values represent the mean ± SEM. Asterisk represents statistically significant differences (**, *p* < 0.01; ***, *p* < 0.001).

**Figure 2 cancers-13-04828-f002:**
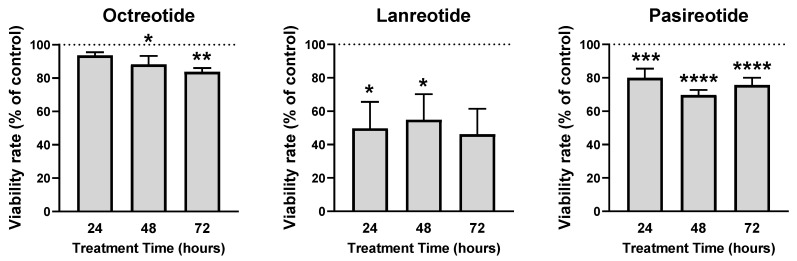
Effect of different somatostatin analogues (octreotide, lanreotide, and pasireotide) on cell proliferation OSS primary cell cultures. Proliferation rate (24- to 72-h treatment) was measured by Alamar Blue reduction. Data are expressed as percent of vehicle-treated control (set at 100%). Values represent the mean ± SEM (*n* = 3–4 tumors, 3–4 replicates/treatment). Asterisk represents statistically significant differences (*, *p* < 0.05; **, *p* < 0.01; ***, *p* < 0.001; ****, *p* < 0.0001).

**Table 1 cancers-13-04828-t001:** In vivo association between SST-subtypes expression in OSCC with overall survival (OS), recurrence rate (RR), and distant metastasis. SSTs expression is expressed as categorical with “>/< median” analysis. The *p*-values were calculated with the log-rank test for the analysis between SSTR >/< median analysis and OS, overall RR, local RR, regional RR, local and regional RR, and distant metastasis.

SSTR	OS	RR	Local RR	Regional RR	Local and Regional RR	Distant Metastasis
** *SST_1_* ** **>/< median**	*p = 0.39* *R* − *0.20*	*p = 0.45* *R 0.10*	*p = 0.99* *R* − *0.04*	*p = 0.97* *R* − *0.04*	*p = 0.29* *R* − *0.23*	*p = 0.7* *R* − *0.08*
** *SST_2_* ** **>/< median**	*p = 0.14* *R 0.16*	*p = 0.23* *R* − *0.20*	*p = 0.13* *R* − *0.22*	***p = 0.04 (*−*)*** ***R* − *0.37***	***p = 0.02 (*−*)*** ***R* − *0.43***	***p = 0.07 (*−*)*** ***R* − *0.32***
** *SST_3_* ** **>/< median**	*p = 0.46* *R* − *0.08*	*p = 0.71* *R 0.07*	*p = 0.73* *R* − *0.07*	*p = 0.70* *R 0.07*	*p = 0.65* *R* − *0.08*	*p = 0.53* *R 0.10*
** *SST_4_* ** **>/< median**	*p = 0.78* *R 0.05*	*p = 0.53* *R* − *0.09*	*p = 0.42* *R* − *0.13*	*p = 0.41* *R* − *0.13*	*p = 0.27* *R* − *0.18*	*p = 0.14* *R* − *0.26*
** *SST_5_* ** **>/< median**	*p = 0.38* *R* − *0.20*	*p = 0.38* *R 0.01*	*p = 0.48* *R 0.01*	*p = 0.47* *R – 0.16*	***p = 0.05 (*−*)*** ***R* − *0.38***	*p = 0.76* *R* − *0.06*

(−), negative correlation; (+), positive correlation. The bold is to highlight the significant result of these analysis and different categories of SST receptors.

**Table 2 cancers-13-04828-t002:** In vivo relationship between SST-subtypes expression in OSCC and Staging data. SSTs Numerical expression is expressed as mean ± SD in each category. Nonparametric Kruskal-Wallis and U-Mann Whitney tests are used to analyze the relationship between SSTs expression and Staging data.

SSTR	pT	pT × 2	pN	pN × 4	pN × 2	pN−/pN+
** *SST_1_* ** **Numerical**	*p = 0.37* pT1: 0 pT2: 14,515 ± 10,312 pT3: 318,394 ± 312,755 pT4: 10,161 ± 6971	*p = 0.34* pT1 + pT2: 12,579 ± 8985 pT3 + pT4: 133,414 ± 124,958	*p = 0.74* pN0: 133,837 ± 124,939 pN1: 4819 ± 3000 pN2a: 5003 ± 3128 pN2b: 68,309 ± 67,711 pN2c: 811 ± 811 pN3b: 2544 ± 1910	*p = 0.78* pN0: 133,837 ± 124,939 pN1: 4819 ± 3000 pN2: 24,709 ± pN3: 2544 ± 1910	*p = 0.92* pN0 + pN1: 112,334 ± 104,147 pN2 + pN3: 14,633 ± 12,181	*p = 0.63* pN-: 133,837 ± 124,939 pN+: 12,016 ± 8921
** *SST_2_* ** **Numerical**	*p = 0.80* pT1: 23,645 ± 20,298 pT2: 25,851 ± 7180 pT3: 321,684 ± 305,878 pT4: 24,318 ± 12,041	*p = 0.34* pT1 + pT2: 25,557 ± 6501 pT3 + pT4: 143,264 ± 122,292	***p = 0.02, (*−*)*** pN0: 151,455 ± 121,862 pN1: 12,049 ± 6652 pN2a: 36,269 ± 19,590 pN2b: 41,679 ± 33,279 pN2c: 3003 ± 561 pN3b: 1887 ± 1074	***p = 0.02, (*−*)*** pN0: 151,455 ± 121,862 pN1: 12,049 ± 6652 pN2: 26,984 ± 12,566 pN3: 1887 ± 1074	***p = 0.05, (*−*)*** pN0 + pN1: 128,221 ± 101,693 pN2 + pN3: 15,576 ± 7674	***p = 0.03, (*−*)*** pN−: 151,455 ± 121,862 pN+: 14,636 ± 5792
** *SST_3_* ** **Numerical**	*p = 0.41* pT1: 7051 ± 7051 pT2: 1,520,615 ± 151,915 pT3: 255,331 ± 251,582 pT4: 11,187 ± 9756	*p = 0.19* pT1 + pT2: 1,318,806 ± 131,685 pT3 + pT4: 108,845 ± 100,565	*p = 0.51* pN0: 109,093 ± 100,554 pN1: 2189 ± 1436 pN2a: 2166 ± 1935 pN2b: 9,877,900 ± 987,692 pN2c: 0 pN3b: 1648 ± 1084	*p = 0.13* pN0: 109,093 ± 100,554 pN1: 2189 ± 1436 pN2: 3,293,315 ± 329,228 pN3: 1648 ± 1084	*p = 0.97* pN0 + pN1: 91,275 ± 8342 pN2 + pN3: 1,797,125 ± 179,576	*p = 0.94* pN-: 109,093 ± 100,554 pN+: 1,318,475 ± 1,316,879
** *SST_4_* ** **Numerical**	*p = 0.65* pT1: 0 pT2: 83,614 ± 75,783 pT3: 25,890 ± 25,558 pT4: 11,869 ± 10,693	*p = 0.67* pT1 + pT2: 72,466 ± 65,757 pT3 + pT4: 17,477 ± 11,767	*p = 0.33* pN0: 67,623 ± 49,981 pN1: 1224 ± 495 pN2a: 2012 ± 2012 pN2b: 35,105 ± 33,658 pN2c: 0 pN3b: 989 ± 989	*p = 0.62* pN0: 67,623 ± 49,981 pN1: 1224 ± 495 pN2: 12,372 ± 11,296 pN3: 989 ± 989	*p = 0.87* pN0 + pN1: 56,556 ± 41,789 pN2 + pN3: 7198 ± 6180	*p = 0.65* pN-: 56,556 ± 41,789 pN+: 5605 ± 4529
** *SST_5_* ** **Numerical**	*p = 0.24* pT1: 0 pT2: 5915 ± 2586 pT3: 174,997 ± 173,940 pT4: 17,632 ± 9363	*p = 0.97* pT1 + pT2: 5126 ± 2292 pT3 + pT4: 80,597 ± 69,296	*p = 0.87* pN0: 793,751 ± 69,346 pN1: 11092 ± 10581 pN2a: 4833 ± 1503 pN2b: 7949 ± 6748 pN2c: 1575 ± 1192 pN3b: 5589 ± 4638	*p = 0.77* pN0: 79,3751 ± 69,346 pN1: 11,092 ± 10,581 pN2: 4786 ± 2153 pN3: 5589 ± 4638	*p = 0.58* pN0 + pN1: 67,991 ± 57,803 pN2 + pN3: 5151 ± 2278	*p = 0.63* pN-: 793,751 ± 69,346 pN+: 6735 ± 3100

Abbreviations: pT, tumor size (pT1, pT2, pT3, pT4); pT × 2 (pT1 + pT2/pT3 + pT4); pN, cervical metastasis (pN0/pN1/pN2a/pN2b/pN3); pN × 4 (pN0/pN1/pN2/pN3); pN × 2 (pN0 + pN1/pN2 + pN3), pN− (pN0) vs. pN+ (pN1, pN2, pN3); (−), negative correlation; (+), positive correlation. The bold is to highlight the significant result of these analysis and different categories of SST receptors.

**Table 3 cancers-13-04828-t003:** In vivo relationship between SST-subtypes expression in OSCC and histopathological data. SSTs Categorical expression is expressed as >/< median. Chi2 or Fisher tests are used to analyze the relationship between SSTs expression and histopathological data.

**SSTR**	**pT**	**pT × 2**	**pN**	**pN × 4**	**pN × 2**	**pN−/pN+**	**Stage**	**Stage × 2**
** *SST_1_* ** **>/< median**	*p = 0.46* *R 0.13*	*p = 0.5* *R 0.14*	*p = 0.41* *R − 0.01*	*p = 0.30* *R* − *0.01*	*p = 0.54* *R − 0.04*	*p = 0.44* *R 0.08*	*p = 0.37* *R 0.17*	*p = 0.17* *R 0.22*
** *SST_2_* ** **>/< median**	*p = 0.68* *R − 0.15*	*p = 0.20* *R − 0.19*	***p = 0.04 (*−*)*** ** *R − 0.43* **	***p = 0.05 (*−*)*** ** *R − 0.43* **	***p = 0.08*** *R* − *0.29*	***p = 0.03 (*−*)*** ** *R − 0.38* **	*p = 0.42* *R − 0.19*	*p = 0.11* *R − 0.26*
** *SST_3_* ** **>/< median**	*p = 0.13* *R 0.33*	** *p = 0.03 (+)* ** ** *R 0.38* **	*p = 0.49* *R 0.08*	*p = 0.94* *R 0.08*	*p = 0.63* *R = 0.08*	*p = 0.62* *R 0.08*	*p = 0.16* *R 0.33*	** *p = 0.04 (+)* ** ** *R 0.34* **
** *SST_4_* ** **>/< median**	*p = 0.60* *R 0.17*	*p = 0.36* *R 0.11*	*p = 0.18* *R 0.01*	*p = 0.34* *R 0.01*	*p = 0.53* *R -0.05*	*p = 0.51* *R 0.11*	*p = 0.50* *R 0.20*	*p = 0.28* *R 0.22*
** *SST_5_* ** **>/< median**	*p = 0.36* *R 0.01*	*p = 0.44* *R − 0.08*	*p = 0.79* *R 0.04*	*p = 0.79* *R 0.04*	*p = 0.73* *R 0.08*	*p = 0.73* *R 0.08*	*p = 0.37* *R = 0.07*	*p = 0.59* *R − 0.02*
	**G**	**DOI × 3**	**PTI**	**PTI × 2**	**PNI**	**LVI**	**Invasion Front**	**Uniformity**
** *SST_1_* ** **>/< median**	*p = 0.43* *R 0.08*	*p = 0.17* *R 0.23*	*p = 0.60* *R 0.08*	*p = 0.41* *R 0.09*	*p = 0.71* *R 0.10*	*p = 0.31* *R 0.20*	** *p = 0.08 (+)* ** ** *R 0.31* **	** *p = 0.08 (+)* ** ** *R 0.31* **
** *SST_2_* ** **>/< median**	*p = 0.56* *R − 0.02*	***p = 0.04 (*−*)*** *R − 0.21*	** *p = 0.06* ** *R 0.20*	** *p = 0.08* ** *R 0.32*	*p = 0.57* *R 0.01*	*p = 0.60* *R 0.02*	** *p = 0.08 (+)* ** ** *R 0.31* **	** *p = 0.08 (+)* ** ** *R 0.31* **
** *SST_3_* ** **>/< median**	*p = 0.56* *R − 0.02*	*p = 0.52* *R 0.17*	*p = 0.37* *R 0.13*	*p = 0.58* *R − 0.02*	*p = 0.10* *R − 0.27*	*p = 0.31* *R − 0.14*	*p = 0.30* *R 0.16*	*p = 0.30* *R 0.16*
** *SST_4_* ** **>/< median**	*p = 0.41* *R − 0.09*	*p = 0.79* *R 0.09*	*p = 0.29* *R 0.11*	*p = 0.31* *R 0.19*	*p = 0.15* *R − 0.25*	*p = 0.22* *R − 0.18*	** *p = 0.02 (+)* ** ** *R 0.40* **	** *p = 0.03 (+)* ** ** *R 0.40* **
** *SST_5_* ** **>/< median**	*p = 0.18* *R − 0.25*	*p = 0.86* *R − 0.07*	*p = 0.54* *R 0.02*	*p = 0.73* *R 0.09*	*p = 0.73* *R 0.10*	*p = 0.71* *R 0.08*	** *p = 0.01 (+)* ** ** *R = 0.46* **	** *p = 0.01 (+)* ** ** *R = 0.46* **

Abbreviations: DOI, Depth Of Invasion (>5 mm, 5–10 mm, >10 mm); DFS, Disease Free Survival; G, Grade; Invasion (expansive (+) vs. infiltrative (*−*)); LVI, lymphovascular invasion; OS, overall survival; pN, cervical metastasis (pN0/pN1/pN2a/pN2b/pN3); pN × 4 (pN0/pN1/pN2/pN3); pN × 2 (pN0 + pN1/pN2 + pN3), pN − (pN0) vs. pN + (pN1, pN2, pN3); perineural invasion; pT, tumor size (pT1,pT2,pT3,pT4); pT × 2 (pT1 + pT2/pT3 + pT4); PTI, peritumoral inflammation (mild/moderate/severe) PTI × 2 (absent+mild/moderate + severe); RR recurrence rate; Stage (I/II/III/IV); Stage × 2 (I + II/III+/IV); Invasion front (poor defined tumor edges (*−*) vs. well defined edges (+)); (*−*), negative correlation; (+), positive correlation. The bold is to highlight the significant result of these analysis and different categories of SST receptors.

**Table 4 cancers-13-04828-t004:** In vivo relationship between SST-subtypes expression in OSCC and histopathological factors. SSTs numerical expression is expressed as mean ± SD in each category. Spearman test and Nonparametric Kruskal-Wallis and U-Mann Whitney tests are used to analyze the relationship between SSTs expression and histopathological data.

**SSTR**	**G**	**DOI**	**DOI × 3**	**PTI**	**PTI × 2**
** *SST_1_* ** **Numerical**	*p = 0.73* G1:143400 ± 138998 G2:16222 ± 9026	*p = 0.68* *R − 0.08*	*p = 0.77* <5 mm:0 5–10 mm: 180,295 ± 166,375 > 10 mm: 9534 ± 5349	*p = 0.42* Mild: 2586 ± 1178 Moderate:16,856 ± 156,073 Severe: 8339 ± 4533	*p = 0.90* Abs + mild: 8513 ± 6027 Mod + sev:130,371 ± 118,966
** *SST_2_* ** **Numerical**	*p = 0.96* G1: 154,884 ± 135,804 G2:27,144 ± 9505	*p = 0.42* R *−* 0.08	*p = 0.11* < 5 mm: 14,620 ± 9821 5–10 mm: 190,516 ± 162,368 > 10 mm:20,776 ± 9683	*p = 0.11* Mild: 10,567 ± 3660 Moderate:177,377 ± 152,420 Severe:25,336 ± 14,448	*p = 0.22* Abs + mild: 20,281 ± 10,287 Mod + sev: 141,177 ± 116,175
** *SST_3_* ** **Numerical**	*p = 0.88* G1:114,126 ± 111,900 G2:1,170,866 ± 1,161,515	*p = 0.82* R − 0.13	*p = 0.68* < 5 mm: 1525 ± 3525 5–10 mm: 1,452,796 ± 1,314,132 > 10 mm: 9559 ± 7310	*p = 0.14* Mild: 1133 ± 452 Moderate: 1,363,582 ± 1,232,491 Severe: 1702 ± 1702	*p = 0.75* Abs + mild: 9511 ± 8388 Mod + sev:1,039,325 ± 940,660
** *SST_4_* ** **Numerical**	*p = 0.93* G1:55,744 ± 54,997 G2:25,478 ± 13,957	*p = 0.86* *R − 0.08*	*p = 0.81* < 5 mm: 558 ± 558 5–10 mm:72,668 ± 65,741 > 10 mm:21,517 ± 14620	*p = 0.18* Mild: 629 ± 397 Moderate:19,210 ± 13,131 Severe:198,323 ± 198,093	*p = 0.39* Abs + mild: 9825 ± 9203 Mod + sev: 618,56 ± 47,519
** *SST_5_* ** **Numerical**	***p = 0.02 (*−*)*** G1:86,155 ± 76,907 G2:8097 ± 6496	*p = 0.91* *R* − *0.06*	*p = 0.53* < 5 mm: 2890 ± 2430 5–10 mm:97,734 ± 92,515 > 10 mm:13,180 ± 7217	*p = 0.44* Mild: 6133 ± 3537 Moderate:91,039 ± 86,784 Severe:8202 ± 6380	*p = 0.80* Abs + mild:13,630 ± 8181 Mod + sev:71,315 ± 66,089
	**PNI**	**LVI**	**Invasion front**	**Uniformity**	
** *SST_1_* ** **Numerical**	*p = 0.81* Yes: 106,694 ± 100,039 No:18,963 ± 13,299	*p = 0.75* Yes:139,111 ± 131,580 No:13,367 ± 8440	***p = 0.01 (+)*** Expansive: 448,221 ± 412,110 Infiltrative: 5781 ± 3024	***p = 0.01 (+)*** Uniform: 448,221 ± 412,110 Non-Uniform: 5781 ± 3024	
** *SST_2_* ** **Numerical**	*p = 0.81* Yes: 117,537 ± 97,878 No: 31,020 ± 10,019	*p = 0.80* Yes: 150,406 ± 128,653 No: 24,433 ± 7121	***p = 0.03 (+)*** Expansive: 446,069 ± 403,333 Infiltrative: 19,732 ± 5736	***p = 0.03 (+)*** Uniform: 446,069 ± 403,333 Non-uniform: 19,732 ± 5736	
** *SST_3_* ** **Numerical**	*p = 0.15* Yes: 86,476 ± 80,549 No: 1,979,708 ± 19,75,010	*p = 0.42* Yes: 113,586 ± 105,890 No: 1,237,553 ± 1,234,483	*p = 0.17* Expansive: 3,630,478 ± 3,241,581 Infiltrative: 6073 ± 4069	*p = 0.17* Uniform: 3,630,478 ± 3,241,581 Non-uniform: 6073 ± 4069	
** *SST_4_* ** **Numerical**	*p = 0.15* Yes: 46.034 ± 39.704 No: 28.568 ± 20.697	*p = 0.31* Yes: 8520 ± 6834 No: 79.666 ± 62.136	***p = 0.05 (+)*** Expansive: 177.277 ± 163.059 Infiltrative: 12.858 ± 8194	***p = 0.05, (+)*** Uniform: 177.277 ± 163.059 Non-uniform: 12.858 ± 8194	
** *SST_5_* ** **Numerical**	*p = 0.17* Yes: 66,279 ± 55,467 No: 3146 ± 575	*p = 0.63* Yes: 83,845 ± 72,960 No: 5962 ± 2429	***p < 0.01 (+)*** Expansive: 245,340 ± 28,491 Infiltrative: 7462 ± 40,152	***p < 0.01, (+)*** Uniform: 245,340 ± 28,491 Non-uniform: 7462 ± 40,152	

Abbreviations: G, grade; DOI, Depth of Invasion; DOI × 3 (<5mm, 5–10mm. >10mm); Invasion front (expansive (+) vs. infiltrative (*−*)); PTI (mild, moderate, severe), PTI × 2 (absent + mild/moderate+severe); PNI, perineural invasion; LVI, lymphovascular invasion; Uniformity (poor defined tumor edges (*−*) vs. well defined edges (+)); (*−*), negative correlation; (+), positive correlation. The bold is to highlight the significant result of these analysis and different categories of SST receptors.

**Table 5 cancers-13-04828-t005:** In vivo relationship between SST-subtypes expression in OSCC and lymph node pathological data. Spearman correlation test was used for the analysis between SSTs numerical expression and lymph node results.

SSTR	N° + lymph	N° ENE +	Size (mm)
** *SST_1_* ** **Numerical**	*p = 0.68* *R 0.07*	*p = 0.83* *R. 0.04*	*p = 0.29* *R 0.20*
** *SST_2_* ** **Numerical**	***p < 0.01 (*−*)*** *R − 0.535*	***p = 0.03 (*−*)*** *R − 0.40*	***p = 0.05 (*−*)*** *R − 0.36*
** *SST_3_* ** **Numerical**	*p = 0.63* *R 0.09*	*p = 0.96* *R* − *0.01*	*p = 0.62* *R 0.09*
** *SST_4_* ** **Numerical**	*p = 0.88* *R 0.02*	*p = 0.64* *R − 0.08*	*p = 0.28* *R 0.20*
** *SST_5_* ** **Numerical**	*p = 0.94* *R 0.01*	*p = 0.84* *R 0.03*	*p = 0.89* *R 0.03*

Abbreviations: ENE+, extranodal extension; (*−*), negative correlation; (+), positive correlation. The bold is to highlight the significant result of these analysis and different categories of SST receptors.

## Data Availability

The data presented in this study are available on request from the corresponding authors.

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
