# Peer review of "Molecular and Clinical Implications of Somatostatin Receptor Profile and Somatostatin Analogues Treatment in Oral Cavity Squamous Cell Carcinoma"

_cancers, 2021, doi:10.3390/cancers13194828_

Round 1

Reviewer 1 Report

Dear Authors, 

The papers you presented is clear and well written. The study is well explained and the results clearly presented. However, the small number of patients examined do not allow to strengthen the power of your results. Moreover, the follow up regimen should be consider larger than 5 years to give soundness to the presented results. I also like to underline that low stage - low grade OSCC do not need adjuvant treatment and the suggested strategy to administer SSA in these patients does not seem to be a good alternative strategy.

Due to these considerations, a major revision is at least to be considered

Author Response

A point-by-point response to the comments

We sincerely thank the Editor for allowing us to resubmit a thoroughly revised version of our manuscript to respond and address the Reviewers' concerns. We also thank the reviewers for their constructive comments, which we found very helpful for improving the quality of our article. Accordingly, specific changes have been made in the manuscript, based on these comments, as described in detail below in a point-by-point description of the changes introduced and how Reviewers’ concerns were addressed. Changes within the manuscript are highlighted in red. We hope that the responses and provided modifications will appropriately satisfy the Reviewers, and therefore, our manuscript can be accepted for publication in Cancers.

RESPONSES TO REVIEWER #1:

  1. Reviewer: The paper you presented is clear and well written. The study is well explained and the results clearly presented. However, the small number of patients examined do not allow to strengthen the power of your results. Moreover, the follow up regimen should be considered larger than 5 years to give soundness to the presented results

Authors: We thank the Reviewer for the laudatory comments and this observation. We understand that a great number of patients and a 5-year follow-up period might be desirable to unveil the full potential of the overall survival results. Accordingly, following the Reviewer’s suggestion, we have included a sentence in the last paragraph of the study indicating this limitation (page 12, lines 418-422). “The present study has some limitations: the limited number of cases analyzed and the short follow-up time for the analysis of the OS, which is usually measured on a 5-years period. Therefore, we will continue increasing the number of samples for future investigations and continue following these patients for a proper analysis of the impact of SSTs on patient´s survival”. However, these planned studies will necessarily require several years (and associated extra funding) and are meant to be the subject of independent manuscripts in the future. In this sense, although this study cannot assure that SST has a true impact on the overall survival, we want to emphasize the novel results included in the manuscript demonstrating that the high levels of expression of some SST-subtypes in OSCC (i.e. SST2) are significantly associated with better prognosis, histopathological tumor features and a lower incidence for recurrence at 2-years, and that the treatment with different somatostatin analogues can significantly decreased proliferation rate of OSCC cells. Accordingly, we would respectfully request to the Reviewer and Editor this additional and challenging data not to be considered as a requisite for the potential acceptance of our present revised manuscript.

  1. Reviewer: I also like to underline that low stage - low-grade OSCC do not need adjuvant treatment and the suggested strategy to administer SSA in these patients does not seem to be a good alternative strategy.

Authors: We appreciate and understand this constructive criticism of the Reviewer. We apologize for not being more explicit in this aspect. Indeed, low-grade – low stage OSCC are not treated with any adjuvant therapy. Our results showed that SSTs expression is related to histopathological factors present in tumors with a less aggressive histopathological behavior like an expansive vs. infiltrative front of invasion, lower histopathological grade, or uniform edges compared to poorly defined ones. However, SSTs were significantly elevated in all tumor samples compared to the adjacent tissue, especially SST2 and SST3 (p<0.01 and p<0.001). Therefore, based on the Reviewer´s comment, we have now included in the first paragraph of the results sections (section 3) that most of our patients were Stage IV. Consequently, the hypothesis that SSAs exert antitumoral effects on OSCC cells would not be just limited to the low grade – low stage.

Reviewer 2 Report

Dear Authors,

The article: 'Molecular and Clinical Implications of Somatostatin Receptor Profile and Somatostatin Analogues Treatment in Oral Cavity Squamous Cell Carcinoma' was described to : 1) to quantitatively analyze the expression profile of SSTs in OSCC vs. adjacent healthy-tissues obtained within the same patient in a well-characterized cohort of patients; 2) to assess the putative in vivo association between the expression of all SSTs in the tumor and relevant clinical/histopathological data parameters (stage, histological grade, tumor invasion, presence of metastasis, recurrence, overall survival, etc.); and, 3) to explore and compare side by side the direct antitumor effects of different SSAs.

English language and style are fine.
Correct (page 1): Citation: Iemura, K.; Yoshizaki, Y.,
Kuniyasu, K.; Tanaka, K. Attenuated chromosome oscillation as a
cause of chromosomal instability in
cancer cells. Cancers 2021, 13, x

Information on the consent of the bioethical commission should be included in the materials and methods. Add information about TNM classification.

Punctuation and editorial errors in the text should be corrected.

line 149: 2.4. Primary OASS Culture
slip to page 4

p value and vs should be written italics.

number - rounding to the same digit after the point.

The quality of the figures is too low (e.g. Figure 1, 2).

The description of table 9 should be together with the content of the table.

The table with the list of abbreviations is missing,

Incorrect citation record type at reference point. 
Chi, A.C., T.A. Day, and B.W. Neville, Oral cavity and oropharyngeal squamous cell carcinoma--an update. CA Cancer J Clin, 2015. 441
65(5): p. 401-21.
vs. 
Zhang, J.; Yu, X.; Guo, P.; Firrman, J.; Pouchnik, D.; Diao, Y.; Samulski, R.J.; Xiao, W. Satellite Subgenomic Particles Are Key Regulators of Adeno-Associated Virus Life Cycle. Viruses 202113, 1185.

To sum up, article can be accepted after major revision.

Author Response

A point-by-point response to the comments

We sincerely thank the Editor for allowing us to resubmit a thoroughly revised version of our manuscript to respond and address the Reviewers' concerns. We also thank the reviewers for their constructive comments, which we found very helpful for improving the quality of our article. Accordingly, specific changes have been made in the manuscript, based on these comments, as described in detail below in a point-by-point description of the changes introduced and how Reviewers’ concerns were addressed. Changes within the manuscript are highlighted in red. We hope that the responses and provided modifications will appropriately satisfy the Reviewers, and therefore, our manuscript can be accepted for publication in Cancers.

RESPONSES TO REVIEWER #2:

The article: 'Molecular and Clinical Implications of Somatostatin Receptor Profile and Somatostatin Analogues Treatment in Oral Cavity Squamous Cell Carcinoma' was described to: 1) to quantitatively analyze the expression profile of SSTs in OSCC vs. adjacent healthy-tissues obtained within the same patient in a well-characterized cohort of patients; 2) to assess the putative in vivo association between the expression of all SSTs in the tumor and relevant clinical/histopathological data parameters (stage, histological grade, tumor invasion, presence of metastasis, recurrence, overall survival, etc.); and, 3) to explore and compare side by side the direct antitumor effects of different SSAs. English language and style are fine.

  1. Reviewer: Correct (page 1): Citation: Iemura, K.; Yoshizaki, Y., Kuniyasu, K.; Tanaka, K. Attenuated chromosome oscillation as a cause of chromosomal instability in cancer cells. Cancers 2021, 13, x

Authors: We appreciate this helpful comment of the Reviewer. Following the Reviewer's request, this has been corrected.

  1. Reviewer: Information on the consent of the bioethical commission should be included in the materials and methods.

Authors: We sincerely appreciate this pertinent and important observation and apologize for not mentioning this in the paper before. This information has been now included (page 2, lines 90-92), as follows: The Ethics Committee of the Reina Sofia University Hospital (Cordoba, Spain) approved the study, which was conducted in accordance with the Declaration of Helsinki and with national and international guidelines and approved by the Ethics Committee of the Reina Sofia University Hospital (Cordoba, Spain, Approval # 70180004). Written informed consent was obtained from all the patients

  1. Reviewer: Add information about TNM classification.

Authors: We thank the Reviewer for this critical observation. In this regard, and following the reviewer´s request, we have now included a paragraph at the beginning of the result section explaining in detail the TNM results for Stage, pT, and pN for the study cohort (page 5, lines 212-216), as follows: “Our cohort is comprised by 50% of patients with advanced Stages IV, 16% with Stage III, 29% Stage II, and 5% with Stage I. 34% of our patients belonged to pT4 tumors, 23% were pT3, 37% were pT2 and 5% were pT1. The cervical lymph node involvement was positive in 42% with pN1 in 10%, and with pN2 and pN3 both in 16%”.

  1. Reviewer: Punctuation and editorial errors in the text should be corrected 1) line 149: 2.4. Primary OASS Culture; 2) p-value and vs. should be written italics; 3) number - rounding to the same digit after the point.

Authors: We appreciate these very helpful corrections, which have been corrected in the revised version of the manuscript.

  1. Reviewer: The quality of the figures is too low (e.g. Figure 1, 2).

Authors: We thank the Reviewer for this pertinent comment. We have improved the quality of the graphics and increased the size of the letters so they can be better read (pages 5 and 10). 

  1. Reviewer: The description of table 9 should be together with the content of the table.

Authors: Regarding this comment, we fail to understand what the reviewer means by Table 9 since only 5 tables were included in the manuscript. Nonetheless, we have revised all the tables to make sure that all of them have the correct information included in the legend.

  1. Reviewer: The table with the list of abbreviations is missing.

Authors: We apologize for the error, which has now been corrected and abbreviations list has been included at the end of the manuscript (page 13, line 438).

  1. Reviewer: Incorrect citation record type at reference point.

Authors: We sincerely appreciate this pertinent observation and apologize for not complying with the bibliography style. This comment has been addressed. We have changed all the references to the given sample and highlighted them in red.

Round 2

Reviewer 2 Report

Dear Authors,

Thank you for your corrections. 

The references list need corrections.